# Biosurfactant-Producing *Bacillus velezensis* PW192 as an Anti-Fungal Biocontrol Agent against *Colletotrichum gloeosporioides* and *Colletotrichum musae*

**DOI:** 10.3390/microorganisms10051017

**Published:** 2022-05-12

**Authors:** Watthanachai Jumpathong, Bungonsiri Intra, Jirayut Euanorasetr, Pagakrong Wanapaisan

**Affiliations:** 1Program on Chemical Sciences, Chulabhorn Graduate Institute, Laksi, Bangkok 10210, Thailand; watthanachai@cgi.ac.th; 2Department of Chemistry, Faculty of Sciences, Chiang Mai University, Chiang Mai 50200, Thailand; 3Mahidol University-Osaka University: Collaborative Research Center for Bioscience and Biotechnology (MU-OU:CRC), Faculty of Science, Mahidol University, Bangkok 10400, Thailand; bungonsiri.int@mahidol.ac.th; 4Department of Biotechnology, Faculty of Science, Mahidol University, Bangkok 10400, Thailand; 5Laboratory of Biotechnological Research for Energy and Bioactive Compounds, Department of Microbiology, Faculty of Science, King Mongkut’s University of Technology Thonburi, Khet Thung Khru, Bangkok 10140, Thailand; jirayut.eua@kmutt.ac.th; 6Department of Microbiology, Faculty of Pharmacy, Mahidol University, 447 Sri-Ayutthaya Road, Ratchathevi, Bangkok 10400, Thailand

**Keywords:** *Bacillus velezensis*, biosurfactant, lipopeptides, antifungal, *Colletotrichum* spp.

## Abstract

In this study, plant-root-associated *Bacillus* species were evaluated as antifungal biocontrol agents by analyzing the production of surface bioactive molecules known as lipopeptide biosurfactants. This study aimed to isolate and characterize antifungal biosurfactant-producing *Bacillus* bacterium. *Bacillus*
*velezensis* PW192 was isolated from the rhizosphere of *Lagerstroemia macrocarpa* var *macrocarpa* and identified based on phylogenetic analysis of the 16S rRNA gene. The biosurfactant was excreted to cultured supernatant and exhibited emulsification power up to 60% and a decrease in surface tension from 72 in distilled water to 21 mN/m. The surface tension properties were stable in a broad range of pH from 6 to 10, in high temperatures up to 100 °C, and in salinities with a NaCl concentration up to 12% (*w*/*v*). Starting from 0.5 mg of acid, precipitated crude biosurfactant exhibited antifungal activity toward Anthracnose, caused by the phytopathogens *Colletotrichum gloeosporioides* and *C. musae*. The chemical structures of the biosurfactant were structurally characterized as lipopeptides fengycin A and fengycin B. The stability of the biosurfactant, as well as the antifungal properties of *B. velezensis* PW192, can potentially make them useful as agricultural biocontrol agents, as well as in other biotechnological applications.

## 1. Introduction

*Bacillus* are spore-forming, ubiquitous microorganisms found in nature, including in soil, water, and extreme terrestrial environments [1]. Bacteria from the *Bacillus* genus are known as factories for the production of biologically-active compounds. They produce a variety of potential enzymes, insecticides, polymers, antibiotics, and surfactants [2]. Some of their products are regarded as harmless and are listed as Generally Recognized as Safe (GRAS).

Biosurfactants are surface-active compounds derived from various microbial sources, including bacteria and fungi [3,4]. They are amphiphilic molecules that comprise both hydrophilic and hydrophobic moieties. The remarkable properties of biosurfactants, such as their biodegradability, biocompatibility, low toxicity, high surface activity, and stability under extreme conditions (temperature, pH, and salinity), have attracted researchers [5]. Cyclic lipopeptides (CLPs) are a class of surfactants produced from *Bacillus*. CLPs are biosynthesized by non-ribosomal peptide synthetases (NRPS), which install fatty acid chains (hydrophobic) to cyclic peptide moieties (hydrophilic) [6]. *B*. *subtilis*, B. *amyloliquefaciens*, *B. pumilus*, and *B. licheniformis* have been reported to produce CLPs such as surfactin, iturin, licenysin, and fengycin [7]. Due to their surface-active properties, CLPs increase the bioavailability of hydrophobic substrates around the plant rhizosphere, leading to the indirect promotion of plant growth due to improvements in agriculture soil properties [8]. *Bacillus*-derived surfactants also have antimicrobial and antifungal abilities that enable the elimination of several plant pathogens. In terms of commercial development, biosurfactants have therefore emerged as biological control agents on a sustainable agricultural basis. 

*Colletotrichum*-caused Anthracnose is a common post-harvest disease in various tropical and subtropical fruits and vegetables that causes more than 50% of loss of the agricultural produce [9]. *C. gloeosporioides* causes severe losses of tropical fruits including papaya, mango, and avocado. *C. musae* is the causal agent of anthracnose in banana fruits [10]. The application of potential biological fungicide agents may be a safe and environmentally-compatible method of disease management. The aims of the present study were to (i) isolate and identify the biosurfactant produced by Bacillus species, (ii) elucidate the biosurfactant stability in different conditions including temperature, pH, and salinity, and (iii) determine the antifungal activity of the biosurfactant against the phytopathogens *Colletotrichum gloeosporioides* and *C. musae*. In addition, the chemical structure of the biosurfactant was characterized.

## 2. Materials and Methods

### 2.1. Screening for Bacillus Producing Biosurfactant 

Soils from the surrounding rhizosphere were collected and preserved at 4 °C until bacteria were isolated. Ten grams of soil was serially diluted with 0.85% of NaCl. The diluents were heated at 80 °C for 10 min to eliminate the vegetative cells. The heat-resistant bacteria were screened on Trypticase soy agar medium (TSA, BD Bacto). After 24 h of incubation, the bacterial isolates were Gram-stained and the cell morphologies were observed under a microscope. Gram-positive, rod-shaped bacterium was further screened to examine biosurfactant production.

Biosurfactant production was screened by cultivating the bacterium in modified biosurfactant production medium from Khondee et al. [11] (1 L consisted of 1 g of glucose, 0.5 g of beef extract, 3.3 g of K_2_HPO_4_, 0.14 g of KH_2_PO_4_, 20 g of glycerol, 2.2 g of NH_4_Cl, 0.2 g of NaNO_3_, 0.1 g of FeSO_4_.7H_2_O, and 0.6 g of MgSO_4_). Single colonies were inoculated into 10 mL of the medium. After 5 days of incubation at 30 °C on a 180 rpm orbital shaking incubator, the supernatant was collected by centrifuging the cultivation medium at 8000 rpm for 5 min. The supernatant was filtered through a 0.45-micron sterile filter. The biosurfactant activity of the cell-free supernatants was examined. 

### 2.2. Determination of Biosurfactant Activity

Biosurfactant activity was determined using surface tension assay and emulsification assay. Surface tension was measured with the pendant drop method using a DM-CE1 instrument (Kyowa Interface Science, Saitama, Japan). Deionized water was used as the control. To determine the emulsification index (E_24_), 3 mL of cultured broth was mixed with an equal volume of hexadecane. After being vortexed for 2 min, the mixture was settled for 24 h at 30 °C and the height of the emulsion layer and the total mixture were measured [12]. The emulsification index (E_24_) was calculated as the percentage of the height of the emulsified layer (mm) divided by the total height of the liquid column (mm). The biosurfactant screening experiments were performed in triplicate.

### 2.3. 16S rRNA Phylogenetic Construction

Genomic DNA of the biosurfactant isolate was extracted using a Wizard genomic DNA purification kit (Promega, Madison, WI, USA). The genomic DNA was used as a DNA template for the amplification of the 16S rRNA gene. Full-length 16S rRNA was amplified using 27F (5′-AGAGTTTGATCMTGGCTCAG-3′) and 1492R (5′-TACGGYTACCTTGTTACGACTT-3’) as the forward and reverse primers, respectively. DNA sequencing was conducted using *785F* (5′-GGATTAGATACCCTGGTA-3′) and 907R (5′-CCGTCAATTCMTTTRAGTTT-3′), which are the inter-primers, to identify bacteria. Full-length 16S rRNA was used for phylogenetic construction. The phylogenetic tree was analyzed.

The values of sequence similarities between the biosurfactant isolate (PW192) and available reference strains were computed with the EzBioCloud server (http://eztaxon-e.ezbiocloud.net, accessed on 30 November 2021) [13]. An almost full-length size of the isolate PW192 16S rRNA gene (1497 bp) was aligned with multiple sequences of available *Bacillus*-type strains using CLUSTAL_X software [14]. Evolutionary trees of the taxa were constructed with the neighbor-joining method [15] using the MEGA 7 software [16]. Bootstrap analysis with 1000 replicates was performed to determine the robustness of the tree topologies [17]; only the bootstrap values of 50% or above are shown.

### 2.4. Biosurfactant Production and Extraction

*Bacillus* isolates were cultivated in biosurfactant production medium as mentioned above. The cell-free supernatant was collected after 5 days of incubation via centrifugation at 8000 rpm for 10 min. Biosurfactant was precipitated by adjusting the pH of the supernatant to pH 2 using 1 M hydrochloric acid. The precipitate was washed twice with sterile water and neutralized with 2 N NaOH [18]. The precipitate was then lyophilized to obtain crude biosurfactant in the form of pale-yellow powder.

### 2.5. Stability Analysis of the Biosurfactant

The stability of the biosurfactant in different conditions, including different salinity concentrations, temperatures, and pH, was elucidated. The salinity of the supernatant was adjusted by adding NaCl in concentrations of 4%, 8%, 12% 16%, and 20% (*w*/*v*). The pH of the supernatant was adjusted from 2 to 12 using HCl or NaOH. The effect of temperature stability was determined by incubating the supernatant 4, 25, 40, 60, 80, and 100 °C. The supernatant was incubated for 1 h for the temperature tests and 24 h for the salinity and pH stability tests. The stability of the biosurfactant was determined using surface tension and E24 measurements. All of the experiments were performed in triplicate. All of the triplicate results were expressed as the mean ± standard deviation.

### 2.6. Antagonistic Activity Testing against Plant Pathogens

The antifungal activity of the crude extract was determined using the disc diffusion test. The tested fungi which caused anthracnose disease were *C*. *gloeosporioides* DOA c1060 and *C*. *musae* BCC 13080. The plant pathogenic fungi were obtained from the Department of Agriculture Culture Collection, Thailand. They were cultured on potato dextrose agar (PDA; HiMedia Laboratories, Maharashtra, India) at room temperature for 7 days. After that, the fungal agar blocks were prepared using a sterile cork borer and were transferred to the center of a PDA plate using a sterile needle. The freeze-dried crude biosurfactant sample was dissolved in dimethyl sulfoxide (DMSO) to the concentration of 50 mg/mL, passed through a 0.45-micron filter, and two-fold serial diluted. For each filtered sample, ten microliters was loaded in a sterile paper disc. After the discs were dried in a biosafety cabinet, they were transferred to the PDA plate about 2.8 cm away from the fungal cork. The plate was incubated at room temperature for 7 days, and the size of the inhibition zone was recorded by measuring the distance between the edges of the disc to the edge of fungal colony. As the negative control, 100% DMSO was used. The test was performed in two replicates, and the average size of the inhibition zones was recorded.

### 2.7. Biosurfactant Fractionation

The crude powder was extracted with methanol. The soluble portion was decanted and then evaporated in vacuo to obtain the honey-brown-colored crude biosurfactant. The crude biosurfactant was re-dissolved in 5.0 mL of 1:1 isopropanol/water. The reconstituted crude biosurfactant was then fractionated with preparative reversed phase HPLC using a PFP HPLC column (Luna 5 um Phenomenex PFP column (250 mm × 21.2 mm) connected with an Agilent 1260 Technology HPLC system. The mobile phases consisted of water (solvent A) and isopropanol (solvent B). The gradient of the mobile phases with the flow rate of 10 mL/min was programmed as follows: the isocratic elution of 1%B for 5 min; the gradient elution of 1%B to 100%B for 100 min; the isocratic elution of 100%B for 10 min (column washing); the isocratic elution of 1%B for 15 min (column equilibration for the next fractionation). The eluted fractions were manually collected and divided into five fractions. Each fraction was concentrated under vacuum and then subjected to mass spectrometric analysis, which is described below.

### 2.8. Structural Characterization Using LC–MS/MS

Each collected fraction was subjected to LC–MS/MS analysis using a Dionex HPLC system (Thermo Fisher Scientific, Waltham, MA, USA) coupled with the Q-Exactive Focus mass spectrometer (Thermo Fisher Scientific). Each sample (20 uL) was injected into a C18 analytical column (Acclaim PolarAdvantageII, 3 µm 120Å, 3 × 250 mm, ThermoScientific) and eluted using 0.1% formic acid in water (solvent A) and 0.1% formic acid in acetonitrile (solvent B). The solvent system was set up with the flow rate of 0.300 mL/min and programmed as follows: the isocratic elution of 1%B for 5 min; the linear gradient of 1%B to 55%B for 50 min; the isocratic elution of 100%B for 10 min (column washing); the isocratic elution of 1%B for 15 min (column equilibration for the next analysis). The eluted peaks from each sample were ionized with an electrospray ionization (ESI) source, which underwent MS/MS fragmentation via higher-energy collision dissociation (HCD) under the positive ion mode of data-dependent acquisition (DDA) using these parameters: sheath gas 25, auxiliary gas 10, spray voltage 3.25 kV, capillary temperature 325 °C, S-lens RF 50, and auxiliary gas temperature 200 °C. The parent and fragmented ions were monitored using the Orbitrap mass analyzer with resolutions of 35,000 and 17,000 for the parent and fragmented ions, respectively; the precursor mass range (*m*/*z*) was 400–2000. The post-acquisition analysis was performed using Compound Discoverer 3.1 (Thermo Fisher Scientific).

## 3. Results

### 3.1. Biosurfactant-Producing Bacillus Isolation and Identification

*Bacillus velezensis* PW192 was isolated from the soil sample taken from the rhizosphere of *Lagerstroemia macrocarpa* var *macrocarpa* in the Nakornsawan province of Thailand. PW192 is a Gram-positive, motile, and rod-shaped bacterium. The strain was identified via 16S rRNA gene sequence analysis. Based on genetic data and phylogenetic tree construction, the PW192 was designated as *Bacillus velezensis*, of which, the sequence showed the highest similarity to that of strain *Bacillus velezensis* CR-502^T^ (99.9%). Furthermore, phylogenetic analysis showed that isolate PW192 clustered with *B. velezensis* CR-502^T^*, B. siamensis* KCTC 13613^T^ (99.8%), and *B. amyloliquefaciens* DSM 7^T^ (99.5%) (Figure 1). Based on genetic data and phylogenetic tree construction (Figure 1), PW192 was designated as *Bacillus velezensis*. 

### 3.2. Property of Biosurfactant 

To determine the level of biosurfactant production, PW192 was grown as described in the Materials and Methods section. After 5 days of incubation, the cell-free supernatant was collected and its biosurfactant activity was examined. The supernatant could reduce the surface tension from 74 (deionized water) to 21 mN/m, and its emulsification index was determined to be 60.17%. 

The stability of the biosurfactant was also tested under different temperatures, pH, and salt concentrations. The surface tension activities and E24 of the supernatant were measured after the supernatant was treated with different conditions. The supernatant was incubated at 4 to 100 °C for 1 h before measurement. Overall, the surface tension activities and E24 of the supernatant were maintained in all of the tested temperatures (Figure 2a). The pH of the supernatant was adjusted to the range of 2–12 and maintained for 24 h. The surface tension activities and E24 were stabilized at a pH ranging from 6–10, of which pH 6 showed the highest surface tension activity of 21.70 mN/m (Figure 2b). The salt resistance of the biosurfactant was determined by adding 0–20% (*w*/*v*) NaCl into the supernatant and maintaining this condition at room temperature for 24 h. The supernatant-containing biosurfactant was stable in 0–12% (*w*/*v*) NaCl concentrations (Figure 2c).

### 3.3. Antagonistic Activity against Anthracnose-Causing Pathogen 

To evaluate the potential bioactivity of the biosurfactant samples, the antifungal activity against the causative agents of anthracnose disease in mango and chili, *C*. *gloeosporioides* c1060 and *C*. *musae* BCC 13080, was determined using the disc diffusion susceptibility test [19]. The result showed that an initial amount of 500 micrograms of sample could inhibit both fungi (*C. gloeosporioides* c1060 and *C*. *musae* BCC 13080), as shown in Table 1 and Figure 3. The antifungal activity was confirmed from the absence of activity from DMSO.

### 3.4. LC–MS/MS Analysis of the Fractionated Biosurfactants 

To identify the unknown structures of lipopeptides biosynthesized by *B. velezensis*, data-dependent acquisition (DDA) LC–MS/MS, a top-down approach commonly used for peptide characterization, was applied to analyze the structures of lipopeptides fractionated from the crude biosurfactant using preparative HPLC. Illustrated in Figure 4a and Appendix A, the mass spectra of fengycin A predominantly showed doubly charged species [M + 2H]^2+^ of all the precursors, including *m*/*z* 725.3967, 732.4052, 739.4124, 746.4210, 724.4075, and 731.4156. Due to much higher intensity than their corresponding [M + H]^+^, these doubly-charged ions were used for MS/MS analysis. Depicted in Figure 4b,e and Appendix A, the HCD fragmentation revealed the identical product ions at *m*/*z* 1080.5 and 966.5, which were derived from Glu-Orn-Tyr-Thr-Glu-Ala-Pro-Gln-Tyr-Ile and Orn-Tyr-Thr-Glu-Ala-Pro-Gln-Tyr-Ile, respectively. Due to the mass difference by 14.0156 (-CH_2_ group), the compounds with [M + H]^+^ 1449.7859, 1463.8040, 1477.8178, and 1491.8354 were considered as a series of homologous molecules.

According to the MS spectra shown in Figure 4c and Appendix A, six derivatives of fengycin B were doubly protonated, giving rise to [M + 2H]^2+^ of 739.4119, 746.4205, 753.4281, 760.4357, and 738.4226. After being fragmented by HCD, all six of the compounds exhibited the characteristic pattern of fengycin B fragmentation by displaying the identical daughter ions of *m*/*z* 1108.5 and 994.5 (Appendix A). The *m*/*z* values of 1108.5 and 994.5 were matched with Glu-Orn-Tyr-Thr-Glu-Val-Pro-Gln-Tyr-Ile and Orn-Tyr-Thr-Glu-Val-Pro-Gln-Tyr-Ile, respectively. Again, molecules with homologous series were detected, as evidenced by the mass difference of 14.0156 Da between [M + H]^+^ of 1477.8164, 1491.8343, 1505.8499 and 1519.8665. To determine the chain length of fatty acids linked to fengycins, the *m*/*z* value of [M + H – 1108.566 – Glu]^+^ could be employed for calculating of the number of carbons in the fatty acid chain. The numbers of carbons in fatty acids are shown in Table 2.

## 4. Discussion

*Bacillus velezensis* PW192, a Gram-positive, endospore-forming bacterium, was isolated from the Lagerstroemia macrocarpa var macrocarpa rhizosphere. This strain was able to produce a biosurfactant with 61% emulsification activity and could reduce the surface tension from 74 to 21 mN/m. The 16S rRNA phylogenetic analysis showed that this strain is a member of the genus *Bacillus* and is related to the species *Bacillus velezensis*.

Biosurfactants can be applied in different industrial processes under extreme conditions of temperature, pH, and salinity. The stability of the cell-free supernatant was investigated under such conditions. We found that both the surface tension and emulsification activity were stable in temperature treatments ranging from 0 to 100 °C with surface tension remaining (21.2 to 24.7 mN/m) the same as the control (21.2 mN/m). The emulsification activity was also maintained at the tested temperatures (E24 = 64%). These results agree with other reports that proved that heat treatment does not alter the interfacial properties of biosurfactants from *Bacillus* species [20]. The biosurfactant activity remained stable from pH 6 to 10; however, the activity was decreased at pH 2 and 4. The loss of surface activity was observed in acidic pH because of the precipitation of the lipopeptides. The biosurfactants also remained stable at different ionic strengths or salt concentrations ranging from 0 to 12% (*w*/*v*) NaCl. The stability of the biosurfactant in various conditions strongly suggests that it has potential to be applied in industries.

Many strains of *B. velezensis* have been isolated. Some studies have shown that *B. velezensis* can produce a variety of metabolites that are capable of stimulating plant growth and inhibiting plant pathogens, including antibacterial proteins, lipopeptide antibiotics, polyketides, siderophores, and ammonium [21]. Previous reports demonstrated the antifungal activity of *Bacillus velezensis* strains against several phytopathogenic fungi, including *Fusarium graminearum* [22,23] and *Colletotrichum gloeosporioides Penz* [24]. The mechanisms of the antifungal activity of *Bacillus* species are expected to involve the presence of hydrolytic enzymes, e.g., cellulase and protease [25], as well as other antifungal compounds, e.g., bacillomycin-D [24] and lipopeptides [26]. Examples of secondary metabolites produced by *B. velezensis* include amylocyclicin, bacilysin, bacillomycin-D, bacillibactin, bacillaene, difficidin, fengycin, macrolactin, plantazolicin, and surfactin [27]. Lipopeptide compounds, including surfactin, fengycin, and bacillomycin-D, demonstrate antifungal properties [21,28]. Bacillomycin-D and fengycin exhibit synergistic antifungal activity that inhibits the growth of *Fusarium* *oxysporum* [29]. 

In this study, the antagonistic activity of acid-precipitated lipopeptide compounds from *B.* velezensis PW192 against Antracnose disease were studied. The antifungal activity shown by the lipopeptide extract of PW192 suggests it could potentially be used as a biocontrol agent. The potential of a particular lipopeptide to exhibit antimicrobial properties largely depends on its molecular structure. Based on structural characterization using LC–MS/MS, it was clear that *B. velezensis* PW192 produces fengycin A and fengycin B biosurfactants. However, the underlying mechanism for this *Bacillus velezensis* PW192 needs to be investigated further.

## 5. Conclusions

The present study indicated that *B. velezensis* PW192 generates potent biocontrol agents, namely fengycin A and B, that serve as fungicides of *Colletotrichum gloeosporioides* and *Colletotrichum musae*. Moreover, the biosurfactants produced by *B. velezensis* PW192 have high potential for industrial applications, mainly due to their stability when subjected to different environmental conditions. 

## Figures and Tables

**Figure 1 microorganisms-10-01017-f001:**
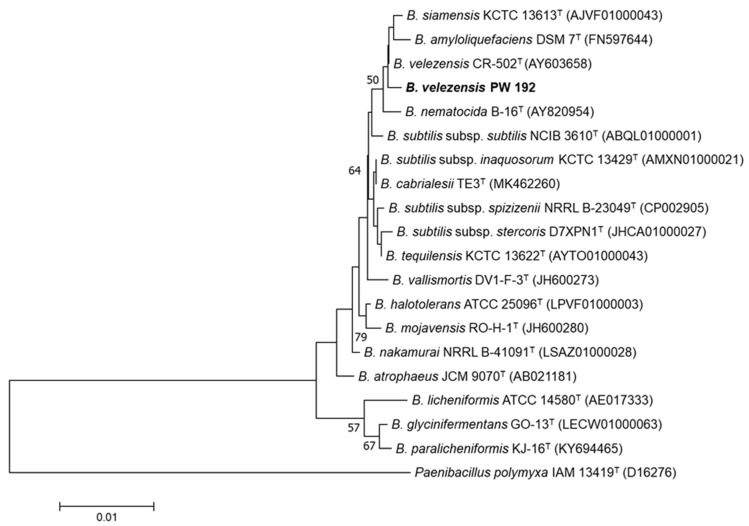
Neighbor-joining phylogenetic tree of *B. velezensis* PW192; the closely-related *Bacillus* spp. Paenibacillus polymyxa IAM 13419^T^ (D16276) was used as the outgroup. Bootstrap values above 50% or higher are shown at branch points based on 1000 resamplings. The scale bar represents 0.01 substitutions per nucleotide position.

**Figure 2 microorganisms-10-01017-f002:**
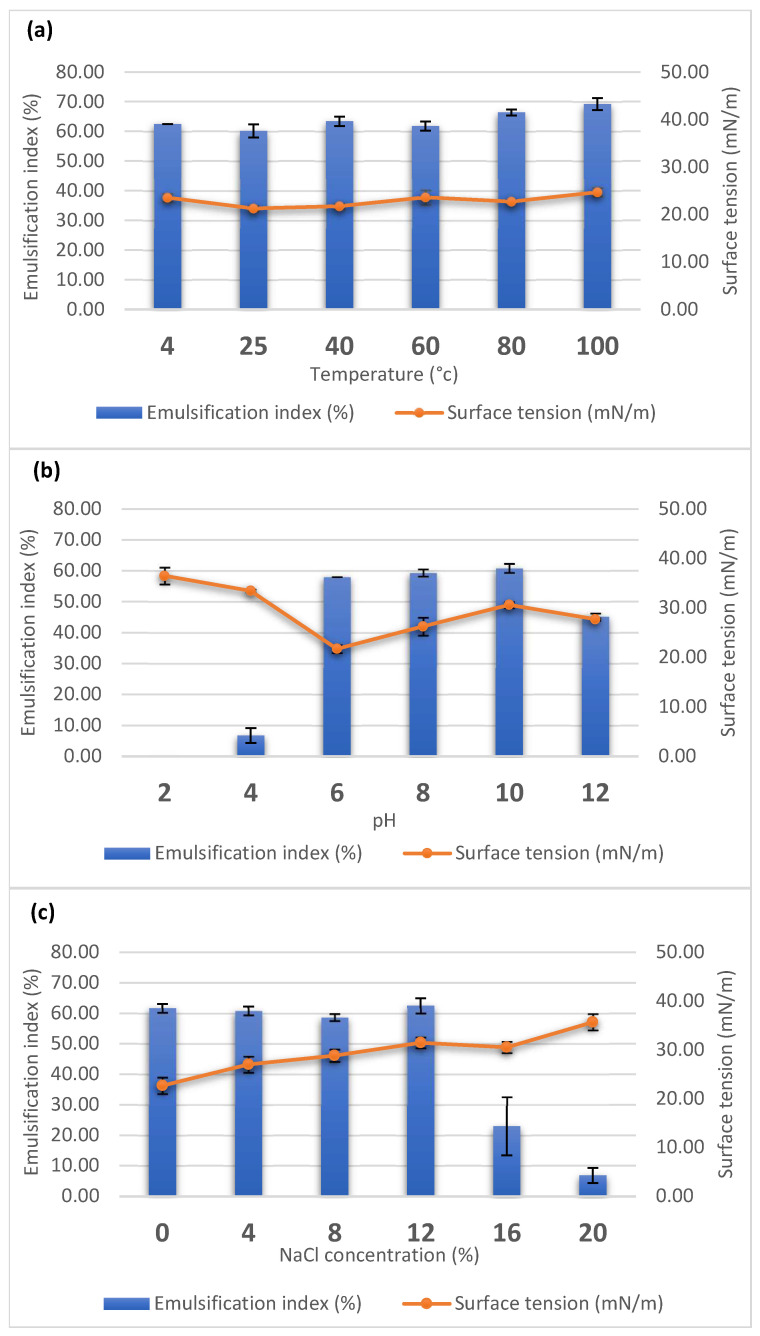
Effect of temperature (**a**), pH (**b**), and NaCl concentration (**c**) on the emulsifying index (%) and surface tension (mN/m) of PW192 biosurfactant. The bar and line plots represent emulsification index and surface tension, respectively. Data presented are the average of triplicate experiments, and error bars indicate standard deviation.

**Figure 3 microorganisms-10-01017-f003:**
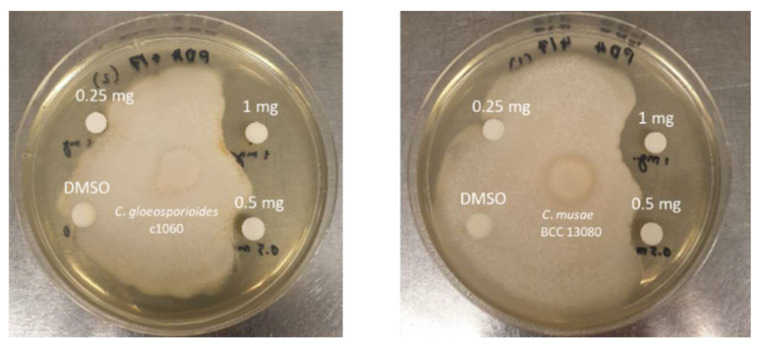
Inhibition zone of the biosurfactant sample against *Colletotrichum* sp.

**Figure 4 microorganisms-10-01017-f004:**
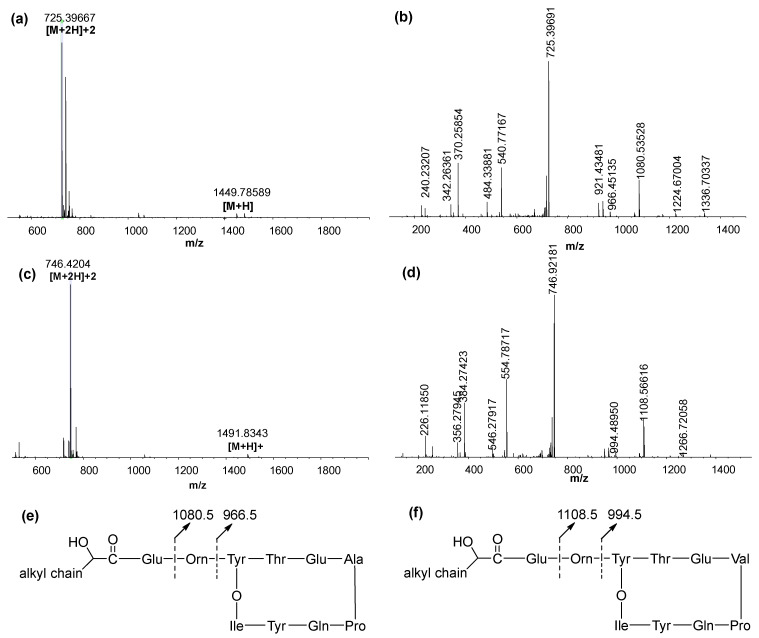
(**a**,**b**) The MS and MS/MS spectra of a fengycin A derivative from [M + H]^+^ 1449.7859. (**c**,**d**) The MS and MS/MS spectra of a fengycin B derivative from [M + H]^+^ 1419.8343. (**e**,**f**) Illustration of the identical product ions derived from fengycin A and fengycin B, respectively.

**Table 1 microorganisms-10-01017-t001:** Inhibition zone of the biosurfactant sample against *Colletotrichum* sp. in milliliters.

	*C. gloeosporioides* c1060	*C. musae* BCC 13080
1 mg	7.5	6.5
0.5 mg	3	3
0.25 mg	0	0
100% DMSO	0	0

Note: The size of inhibition zone was the average of duplicates.

**Table 2 microorganisms-10-01017-t002:** Precursor ions, identical product ions, fatty acid chain length, and fengycin types.

[M + H]^+^	[M + 2H]^2+^	Identical Product Ions	Fatty Acid	Surfactin
1449.7859	725.3967	1080.5, 966.5	15:0	Fengycin A
1463.8040	732.4052	1080.5, 966.5	16:0	
1477.8178	739.4124	1080.5, 966.5	17:0	
1491.8354	746.4210	1080.5, 966.5	18:0	
1447.8081	724.4075	1080.5, 966.5	15:1	
1461.8243	731.4156	1080.5, 966.5	16:1	
1477.8164	739.4119	1108.5, 994.5	15:0	Fengycin B
1491.8343	746.4205	1108.5, 994.5	16:0	
1505.8499	753.4281	1108.5, 994.5	17:0	
1519.8665	760.4357	1108.5, 994.5	18:0	
1461.8242	731.4160	1108.5, 994.5	14:1	
1475.8339	738.4226	1108.5, 994.5	15:1	

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
