# Peer review of "Biosurfactant-Producing Bacillus velezensis PW192 as an Anti-Fungal Biocontrol Agent against Colletotrichum gloeosporioides and Colletotrichum musae"

_microorganisms, 2022, doi:10.3390/microorganisms10051017_

Round 1

Reviewer 1 Report

I will suggest authors to add the reason to do this study in the last paragraph of the introduction. 

Please add the ststtistics details in the method section if used any.

Add more decription of each figures as figure legends.

Please see attached file for more corrections

Author Response

Thank you for your consideration and suggestion. We appreciated to edit the manuscript according to your sugeestion as the following file.

Reviewer 2 Report

see attached file

Review
Row 36. Bacillus genus…. Authors mean “Bacteria from from Bacillus genus ?”
Row 68. Soil from rhizoidsphere? May be- soils from rhizosphere?
Row 71. Trypticase soy agar medium. Give prescription or in the case of commercial medium- producer.
Row 75. Correct “Biosurfactant production was screened”
Row 81. Correct phrase “sterile filter sterile”.
Row 93. Correct 2.316. S rRNA
Row 124. Correct - All experiments
Row 127. The tested fungi, causing anthracnose disease, were consisted of C. gloeosporioides DOA c1060 and C. musae BCC 13080. Please mentioned Culture Collection where they were originated.
Row 157. Correct phrase “Each sample (20 uL) were injected” to was injected
Row 174. May be rhizosphere?
Row 194. Correct to Materials and Methods
Row 286. Correct bengycin to fengycin

Author Response

Thank you for your consideration and comments. We appreciate to edit the manuscript as your suggestion. In adiition, English language of the manuscript was edited by MDPI editing service.

Reviewer 3 Report

The article “Bacillus velezensis PW192 a biosurfactant producing bacterium exhibiting antifungal activity against Colletotrichum gloeosporioides and Colletotrichum musae” is of agronomic relevance.

As general comment the work is well written and designed with relevant results.

In general terms the topic of the article is interesting, the methodology is explicitly presented and the results reported are interesting.

The structure of the paper is correct.

In my opinion, the abstract is too general, please reframe.

The introduction chapter should end with a paragraph indicating the purposefulness of the conducted research. Authors should clearly define the purpose of the work and formulate research hypotheses.

Materials and method section is well described and correspond to the aim set out in the manuscript. 

The tables and figures clearly presenting the obtained results with their appropriate interpretation.

The references are sufficient and necessary.

The paper needs some editorial corrections.

 I recommend the publication of this manuscript in the Microorganisms journal after minor revisions.

Author Response

Thank you for your consideration and comments. We appreciate to edit the manuscript as your suggestion. In addition, English language of the manuscript was edited by MDPI editing service.
